# Multi-Omics Data Integration Reveals Key Variables Contributing to Subgingival Microbiome Dysbiosis-Induced Inflammatory Response in a Hyperglycemic Microenvironment

**DOI:** 10.3390/ijms24108832

**Published:** 2023-05-16

**Authors:** Sarah Lafleur, Antoine Bodein, Joanna Mbuya Malaïka Mutombo, Alban Mathieu, Charles Joly Beauparlant, Xavier Minne, Fatiha Chandad, Arnaud Droit, Vanessa P. Houde

**Affiliations:** 1Oral Ecology Research Group (GREB), Faculty of Dentistry, Université Laval, 2420 rue de la Terrasse, Québec, QC G1V 0A6, Canada; 2Molecular Medicine Department, CHU de Québec Research Center, Université Laval, Québec, QC G1V 4G2, Canada

**Keywords:** microbiome, dysbiosis, periodontitis, hyperglycemia, inflammation, cytokines, metalloproteases, computational biology, omics integration, metagenomic

## Abstract

Subgingival microbiome dysbiosis promotes the development of periodontitis, an irreversible chronic inflammatory disease associated with metabolic diseases. However, studies regarding the effects of a hyperglycemic microenvironment on host–microbiome interactions and host inflammatory response during periodontitis are still scarce. Here, we investigated the impacts of a hyperglycemic microenvironment on the inflammatory response and transcriptome of a gingival coculture model stimulated with dysbiotic subgingival microbiomes. HGF-1 cells overlaid with U937 macrophage-like cells were stimulated with subgingival microbiomes collected from four healthy donors and four patients with periodontitis. Pro-inflammatory cytokines and matrix metalloproteinases were measured while the coculture RNA was submitted to a microarray analysis. Subgingival microbiomes were submitted to 16s rRNA gene sequencing. Data were analyzed using an advanced multi-omics bioinformatic data integration model. Our results show that the genes *krt76*, *krt27*, *pnma5*, *mansc4*, *rab41*, *thoc6*, *tm6sf2*, and *znf506* as well as the pro-inflammatory cytokines IL-1β, GM-CSF, FGF2, IL-10, the metalloproteinases MMP3 and MMP8, and bacteria from the ASV 105, ASV 211, ASV 299, *Prevotella*, *Campylobacter* and *Fretibacterium* genera are key intercorrelated variables contributing to periodontitis-induced inflammatory response in a hyperglycemic microenvironment. In conclusion, our multi-omics integration analysis unveiled the complex interrelationships involved in the regulation of periodontal inflammation in response to a hyperglycemic microenvironment.

## 1. Introduction

Periodontitis is an irreversible chronic inflammatory disease induced by the infiltration of immune cells into the gingival sulcus and is caused by a dysbiosis of the subgingival microbiome [1]. Untreated, periodontitis leads to destruction of the teeth-supporting tissues, bone resorption and ultimately loss of teeth [2,3]. Tissue destruction is caused by the production of inflammatory mediators, including cytokines and matrix metalloproteinases (MMPs), and generates oxidative stress by the excessive production of free radicals that contribute to the inflaming of their environment [4].

Hyperglycemia, or high blood glucose level, is a common manifestation of type 2 diabetes. Hyperglycemic patients have a higher risk of developing periodontitis than normoglycemic patients [5]. Conversely, periodontitis may affect diabetes outcomes [6], indicating that hyperglycemia and periodontitis are bidirectionally interconnected. The 2009–2012 National Health and Nutrition Examination Survey (NHANES) has established that more than 62% of US adults > 65 years with diabetes also have periodontitis [7,8]. Several studies have thereby demonstrated that subgingival microbiome composition is a risk factor for impaired glucose metabolism and insulin resistance [9,10,11]. Periodontitis is thus considered the sixth complication of diabetes [12]. Consequently, the 2017 classification of periodontal disease now considers systemic diseases a key factor for the diagnosis of periodontitis [13].

Inflammation has been proposed as the key factor linking periodontitis and hyperglycemia [14]. Subgingival microbiome dysbiosis has been associated with increased susceptibility to periodontitis in diabetic patients due to a potentially blunted metabolic and immune response from the host [15]. It has also been proposed that inflammation caused by oral bacteria induces insulin resistance [16,17,18]. In cell culture models, high glucose level is known to modulate inflammation associated with periodontitis. The inflammatory response of human U937 macrophages stimulated with lipopolysaccharides (LPS) isolated from the periodontopathogen *Porphyromonas gingivalis* was enhanced by high glucose concentration [19]. Recently, Zhang et al. have also demonstrated that high glucose level activates the inflammasomes in the gingival epithelium [20]. Additionally, periodontitis-induced inflammation can be influenced by subgingival microbiome dysbiosis [1].

At a molecular level, knowledge of the effects of hyperglycemia on the regulation of periodontitis-associated host inflammatory responses and host–microbiome interactions is still scarce. To better understand this knowledge gap, we investigated the impacts of a hyperglycemic microenvironment on the inflammatory response and gene expression of a gingival fibroblast–macrophage coculture model stimulated with dysbiotic subgingival microbiomes using an advanced multi-omics bioinformatic data integration model. Our multi-omics integration model revealed key variables (amplicon sequence variant (ASV), mRNA and cytokines) contributing to periodontitis-induced inflammatory response in a hyperglycemic microenvironment.

## 2. Results

### 2.1. The Hyperglycemic Microenvironment Impacts Pro-Inflammatory Cytokine and Matrix Metalloproteinase Secretion by a Gingival Coculture Model

To assess the impacts of glycemic microenvironments on pro-inflammatory cytokine and metalloproteinase secretion, we measured their secretion rate in the supernatant of a coculture model stimulated with subgingival microbiomes from healthy donors and from patients with periodontitis. We observed that healthy subgingival microbiomes have the same pro-inflammatory potential as dysbiotic subgingival microbiomes (Figure 1A–F). Interestingly, interleukin (IL) 1β (IL-1β) (Figure 1A) and matrix metalloproteinase 3 (MMP3) (Figure 1D) secretion were increased when the coculture model was stimulated with subgingival microbiomes in a hyperglycemic microenvironment (25 mM glucose) compared with a normoglycemic microenvironment (5 mM glucose). In addition, microbiomes from patients with periodontitis increased IL-6 and MMP8 secretion in 25 mM glucose but not in 5 mM glucose (Figure 1C,E). Finally, the glycemic microenvironment did not have an impact on the secretion profiles of tumor necrosis factor α (TNF-α) (Figure 1B), MMP9 (Figure 1F), granulocyte-macrophage colony-stimulating factor (GM-CSF) (Appendix A), basic fibroblast growth factor (FGF2) (Appendix A) or IL-10 (Appendix A).

### 2.2. Subgingival Microbiome Highlights Enrichment of Particular ASV

Bacterial DNA was extracted from both the healthy and the periodontitis subgingival microbiome samples and was submitted to 16S rRNA gene sequencing. Taxonomic annotation of the ASVs revealed that subgingival microbiomes from patients with periodontitis display a different diversity than the samples from the healthy donors (Figure 2A). The Bray–Curtis dissimilarity index (β-diversity) was calculated using the ASV abundance matrix and visualized via a principal component analysis (PCA). The group annotation (healthy/periodontitis) of the samples shows their separation following the first axis of their PCA (Figure 2B), indicating that the subgingival microbiome samples of healthy donors differ in terms of ASV entities and abundance from the corresponding samples of patients with periodontitis.

Differential analyses between the two experimental groups allowed the identification of 12 ASVs (Figure 2C). Eleven ASVs were more abundant in the subgingival microbiome samples from healthy donors. Of these 11 ASVs, 7 were absent from the periodontitis group (for example ASV 178, ASV 519 and *Serratia* genera). ASV 54, associated with *Prevotella,* was more abundant in the periodontitis group. In addition, we found that ASV 299 was more abundant in the periodontitis group and absent from the healthy microbiomes. Unfortunately, it was not possible to annotate ASV 299. It thus may correspond to an uncultured bacterium.

### 2.3. Multi-Omics Correlations by Computational Biology Analyses Reveal Key Variables Contributing to Periodontitis-Induced Inflammatory Response in a Hyperglycemic Microenvironment

In order to find novel associations or correlations between the project’s 3 omics layers (ASV, mRNA and cytokines) while discriminating between the 2 experimental groups (healthy/periodontitis), a multi-omics data integration model was used. It should be noted that data obtained with the multi-omics integration model should not be compared with individual data. First, an arrow plot was generated (Figure 3A). We found that the subgingival microbiome samples from the healthy group and the subgingival microbiomes from the periodontitis group formed two clusters. Second, variables (or data) that are redundant between themselves were selected to bring more information into the multi-omics system. To further understand the relationship between the mRNA, ASV and cytokine variables, a varplot (correlation circle plot) was generated (Figure 3B). We found that positively correlated variables are grouped together, and that negatively correlated or anti-correlated variables are positioned on opposite sides of the plot origin (opposed quadrants).

Finally, the more important variables contributing to the varplot components have been dissected and are presented in Figure 4. Figure 4A is the projection of the individual dots contributing to the 3-layer omics model component 1 while Figure 4B is the projection of the individual dots contributing to the 3-layer omics model component 2. Our results show that the genes keratin 76 (*krt76*), keratin 27 (*krt27*), paraneoplastic antigen-like protein 5 (*pnma5*), MANSC domain-containing protein 4 (*mansc4*), ras-related protein rab-41 (*rab41*), THO complex subunit 6 (*thoc6*), transmembrane 6 superfamily 2 (*tm6sf2*), and zinc finger protein 506 (*znf506*) as well as the pro-inflammatory cytokines IL-1β, GM-CSF, FGF2, IL-10, the metalloproteinases MMP3 and MMP8, and bacteria from the ASV 105, ASV 211, ASV 299, *Prevotella*, *Campylobacter* and *Fretibacterium* genera are key correlated variables contributing to periodontitis-induced inflammatory response in a hyperglycemic microenvironment.

## 3. Discussion

Hyperglycemia and periodontitis are interconnected. However, knowledge of the molecular mechanisms involved in the regulation of periodontitis-associated host inflammatory response and host–microbiome interactions in a hyperglycemic microenvironment is scarce. This knowledge gap was investigated in a gingival fibroblast–macrophage coculture model stimulated with dysbiotic subgingival microbiomes by using an advanced multi-omics bioinformatic data integration model. Multi-omics technologies are powerful tools to integrate biological data [21]. As of the time of this writing very few studies have used these tools to identify the key variables implicated in subgingival microbiome dysbiosis-induced inflammatory response. Our study, one of the first to use this technology, has comprehensively uncovered new relationships between gingival inflammation and hyperglycemia by the integration of microbiome, pro-inflammatory cytokine secretion profile and transcriptomic data. Nevertheless, a major limitation of our multi-omics data integration was the level of biological complexity. mRNA and pro-inflammatory cytokines have been measured in the coculture cells model following stimulation with subgingival microbiomes from two experimental groups in 5 mM and 25 mM glucose. However, significant results were only observed for cytokine secretion in 25 mM glucose. Three layers of omics data (ASV, mRNA and cytokines) in 25 mM glucose have thus been used in addition to the experimental groups (healthy vs. periodontitis) for the data integration. It would not have been possible to perform the multi-omics integration with both glucose concentrations due to the biological complexity of our experimental design. The second limitation of our study is the low number of participants per group, which impacted the data interpretation.

Though the impacts of hyperglycemia on individuals’ pro-inflammatory cytokine and MMP secretions have been reported [12,14,22,23], our multi-omics integration model highlighted new relationships between genes, pro-inflammatory cytokines and bacteria genera implicated in periodontitis-induced inflammatory response in a hyperglycemic microenvironment. Highlighted genes have been associated with DNA damage response (*znf506*) [24], cancer (*pnma5*) [25] and oral cancer (*krt76*, *rab41*) [26,27,28]. We have also identified *tm6sf2,* which is associated with both the risk and the severity of nonalcoholic fatty liver disease [29] as well as with the regulation of inflammatory responses in atherosclerotic lesions [30]. Notably, none of the highlighted genes revealed by our multi-omics model have so far been involved in the regulation of gingival inflammation. Although MMP8 and MMP3 and IL-1β are important inducers of inflammation in our hyperglycemic coculture model, the multi-omics data integration model showed that FGF2, GM-CFS and IL-10 are key pro-inflammatory cytokines that play a major role in periodontitis-induced inflammatory response in a hyperglycemic microenvironment. FGF2 and IL-10 are known to enhance periodontal tissue regeneration [31] and to decrease gingival inflammation [32], respectively. Their highlighted role during periodontitis-induced inflammation may suggest that they are secreted by the coculture model to counteract pro-inflammatory cytokines and matrix metalloprotease secretion. Our multi-omics data integration model unveiled three ASVs (ASV 105, ASV 211, and ASV 299) that contribute to periodontitis-induced inflammation in a hyperglycemic microenvironment while confirming the role of the known periodontopathogens. *Prevotella*, *Campylobacter* and *Fretibacterium* are well known contributors to microbiome dysbiosis during periodontitis [33]. To our knowledge, unannotated ASV 105, ASV 211 and ASV 299 have never been associated with subgingival microbiome dysbiosis or hyperglycemia microenvironments and may correspond to uncultured bacteria. Advancements in culturomics and metagenomics technics may help to further characterize these ASVs. We can thus propose that subgingival microbiome dysbiosis and hyperglycemic microenvironments bring to light both new bacteria genera and gingival fibroblast genes that can play key roles in the host’s inflammatory response. However, future in vitro studies are needed to confirm this hypothesis.

Recent studies have shown that mature biofilms containing pathogenic bacteria have an increased pro-inflammatory capacity over biofilms that contain commensal bacteria [34,35,36]. Contrary to these studies, we found that microbiomes from healthy persons and from patients with periodontitis have the same pro-inflammatory potential in our coculture model. This discrepancy could be due to the nature or composition of the biofilms studied. For many years, scientists have attempted to build in vitro subgingival biofilms to mimic natural subgingival microbiomes in oral health research, with limited success. Whilst reconstructed biofilms in vitro can be used as multispecies community models [37], they do not offer a 100% match with their inoculum after culture enrichment [38,39]. Discrepancies have also been reported between bacteria isolated from oral clinical samples and their counterpart laboratory strains [40,41], indicating that the culture conditions blunted the virulence factors of these microorganisms and impacted biofilm formation. Nevertheless, these studies have clearly demonstrated the complexity of biofilm pathogenicity.

Reactive oxygen species (ROS) are known contributors to both periodontitis- and type 2 diabetes-induced inflammation [42]. Recently, it has been demonstrated that hyperglycemia exacerbates periodontitis in an ROS dependent manner [43] and that pregnant women with gestational diabetes have a higher level of oxidative markers in the saliva, gingival crevicular fluid and serum than healthy pregnant women [44]. Further work is thereby warranted to study the impacts of oxidative stress in our experimental cell culture model. Finally, periodontitis is controlled with frequent and invasive periodontal therapy (scaling and root planing (SRP)) to remove the subgingival microbiome (dental plaque). Adjunctive periodontal treatments to SRP may be developed to improve periodontal clinical outcomes. Candidates for adjunctive periodontal treatments could be natural molecules. Among these natural molecules, polyphenols, probiotics and parabiotics are interesting as they can be used to reduce bacterial load and to control inflammation induced by oral pathogens [45,46,47,48,49,50,51,52,53]. However, future in vitro work and long-term clinical studies are needed to confirm the antibacterial and anti-inflammatory potentials of these natural molecules.

## 4. Material and Methods

### 4.1. Samples Collection

The project was conducted in accordance with the Declaration of Helsinki and was approved by the Research Ethics Committees of Université Laval (ethics number: 2021–013). Samples of subgingival microbiome were collected from four sex-matched, healthy donors (no more than slight gingivitis and no probing pocket depth or attachment loss ≥ 3 mm) and four patients with untreated, moderate-to-severe periodontitis (at least 1 tooth per quadrant with bleeding on probing, pocket depth ≥ 5 mm, and attachment loss ≥ 4 mm) using sterile curettes. Participants that had consumed antibiotics, probiotics, and prebiotics three months prior to the microbiome collection or that had been diagnosed with systemic/metabolic diseases were excluded. Six subgingival sites (tooth) were collected in reduced transport fluid (RTF) medium [54] per participant and pooled to increase the microbial diversity. Each pooled sample was then divided into two parts for: (1) 16s rRNA gene sequencing and (2) coculture cells stimulation. Prior to the coculture model stimulation, the pooled samples optical density (660 nm) was adjusted to 0.2 and the samples heat-inactivated for 30 min at 70 °C to prevent *Streptococcus* spp. overgrowth during the stimulation period (see Appendix A for a summary of the experimental design).

### 4.2. Coculture Model and Stimulation with the Subgingival Microbiome

Primary human gingival fibroblasts (HGF-1) (ATCC #CRL-2014) were immortalized by retroviral-mediated expression of SV40 large T-antigen [55]. Immortalized HGF-1 were cultivated in DMEM supplemented with 10% heat-inactivated (HI)-FBS, 4 mM L-glutamine and 100 μg/mL penicillin/streptomycin + 0.25 μg/mL amphotericin B at 37 °C in 5% CO_2_. Human U937 monocytes (ATCC CRL-1593.2) [56] were cultivated in RPMI medium supplemented with 10% HI-FBS and 100 ug/mL penicillin/streptomycin + 0.25 ug/mL amphotericin B at 37 °C in 5% CO_2_. Two weeks prior to the experiments, the cells were maintained in culture media containing either 5 mM or 25 mM glucose. The 5 mM glucose mimics a normal glycemic microenvironment while the 25 mM represents a hyperglycemic microenvironment.

The coculture model was composed of immortalized HGF-1 overlaid with U937-macrophage-like cells. An arbitrary ratio of 10:1 macrophages to fibroblasts was used. U937 monocytes were differentiated into macrophage-like cells in the presence of 50 ng/mL of phorbol 12-myristate 13-acetate (PMA, Sigma-Aldrich, Oakville, ON, Canada) for 72 h. Macrophage-like cells were then washed to remove non-adherence cells, harvested by scraping, suspended in their culture medium and placed over the immortalized HGF-1. The coculture model was incubated overnight at 37 °C in 5% CO_2_ to allow macrophage adhesion prior to stimulation. The subgingival microbiome samples were used to stimulate the gingival coculture model cultivated either in 5 mM or 25 mM glucose concentration for 24 h. Untreated coculture models served as negative controls.

### 4.3. Pro-Inflammatory Cytokines and Matrix Metalloproteinases Measurement

After stimulation with the oral microbiome for 24 h, the coculture model supernatants were collected, the bacteria in the supernatants removed by centrifugation and the pro-inflammatory cytokines and metalloproteinases were then measured by a Luminex assay (R&D Systems, Toronto, ON, Canada). The cells were washed, lysed and the proteins quantified to normalize the results. Experiments were performed for each collected microbiome sample in two technical replicates for each condition. Results were analyzed using two-way ANOVA test in combination with the Tukey *post hoc* test to assign statistical significance (*p* < 0.05).

### 4.4. 16s rRNA Gene Sequencing

Bacterial DNA from each pooled microbiome sample was extracted by using the ZymoBIOMICS extraction kit from Zymo Research (Irvine, CA, USA). The concentration of DNA was determined with a Nanodrop device (ThermoFisher, Mississauga, ON, Canada). The DNA sequencing (16s rRNA gene sequencing) was performed by the Centre d’expertise et de services Génome Québec (Montreal, QC, Canada) on an Illumina MiSeq apparatus using v3–v4 primers (PE 300 bp (100,000 reads)). The DNA sequencing data were deposited in the SRA database server under accession number PRJNA912124.

### 4.5. RNA Microarray

After stimulation with the oral microbiome for 24 h, the cells were washed with ice-cold PBS and the RNA extracted from the coculture models using the Direct-zol RNA extraction kit from Zymo Research. The concentration of RNA was determined with a Nanodrop device. The RNA sequencing was performed using human Clariom S Affymetrix gene array chips (ThermoFisher, Mississauga, ON, Canada) by the Centre d’expertise et de services Génome Québec. The microarray data were deposited in the GEO database server under accession number GSE222883. Experiments were performed one time in a single biological replicate for each experimental condition.

### 4.6. Bioinformatic Analyses and Data Integration

#### 4.6.1. 16s rRNA ASV Analysis Method

A mean of 32 945 16S sequences were analyzed with an amplicon sequence variant (ASV) methodology using DADA2 pipeline [57]. Briefly, the 16s amplicons were trimmed for their adapters using cutadapt [58] and the options (-m 270 -M 330-discard-untrimmed). The remaining reads were processed using DADA2 commands [57]. The dada2 pipeline allowed us to filter and trim the data based on their quality profile and to remove chimera using an error model based on the sequenced data. A count table of the retrieved ASVs was achieved using the makeSequenceTable function.

#### 4.6.2. Taxonomic Annotation of ASVs

The ASVs were analyzed using Blastn v2.4.0 against RefSeq nt and only hits with a *p* value < 10^−5^ and a percentage of similarity > 95 were considered correct. Remaining ASVs were left with no taxonomy association. The analysis and data representation were produced using R4.2.2. *Pseudomonas* and related ASVs were discarded from the data because they were considered to be contaminants and not a member of the oral subgingival microbiome. ASV counts were normalized using the total annotation depth with the decostand function of the vegan package (v2.6–4) in order to produce relative abundance graphs. ASV counts were also obtained using the DESeq2 R package in order to normalize counts and produce a biplot using the prcomp function of the stats package.

#### 4.6.3. Microarray Analysis

The affymetrix clariom S human CEL files were analyzed in R v4.0.3 [59]. CEL files were imported and normalized using the robust multichip average (RMA) methodology implemented in the oligo v1.54.1 package [60]. The probes were annotated using the affycoretools v1.62.0 [61] and the pd.clariom.s.human v3.14.1 packages [62]. The expression value of genes associated with multiple probes was calculated as the mean of the expression of all the probes targeting it.

#### 4.6.4. Data Integration (16s, Microarray, Cytokines in Hyperglycemic Microenvironment)

Multi-omics data integration was performed using mixOmics R package [63]. Prior to integration, a log2 transformation and centered log ratio (CLR) were applied to the cytokine and ASV data, respectively. Then, a multiblock sparse PLSDA was performed on the microarray and the 16s gene sequencing and cytokine data with 2 principal components were used to identify similar expression profiles within the experimental groups. Finally, we performed a feature selection step to identify the most discriminant profiles between experimental groups; 10 genes and 25 ASV were selected for the first component and 5 genes and 20 ASV were selected for the second component. No selection was performed on the cytokine data. A circle correlations plot was extracted as well as various graphical outputs to depict correlations between molecular features.

## 5. Conclusions

In conclusion, our multi-omics integration analysis unveiled unique interrelated bacterial genera, genes and pro-inflammatory cytokines involved in the regulation of the inflammatory response in a hyperglycemic microenvironment. These data highlight the importance of considering hyperglycemic conditions in the management or development of new treatments for periodontal disease linked with type 2 diabetes. An interesting potential research avenue could be the development of adjunctive periodontal treatments based on natural molecules such as polyphenols, probiotics and parabiotics to reduce bacterial load and to control inflammation induced by oral pathogens.

## Figures and Tables

**Figure 1 ijms-24-08832-f001:**
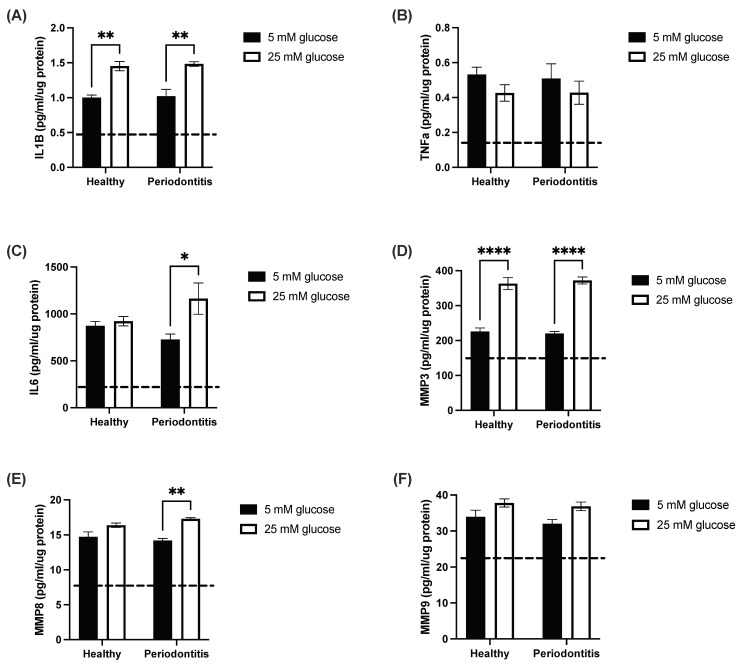
The hyperglycemic microenvironment stimulates pro-inflammatory cytokine and matrix metalloproteinase secretion in a gingival coculture model. Pro-inflammatory cytokine and matrix metalloproteinase secretion of (**A**) IL-1β, (**B**) TNF-α, (**C**) IL-6, (**D**) MMP3, (**E**) MMP8 and (**F**) MMP9 by the coculture model following 24-h stimulation with subgingival microbiomes from healthy participants (healthy) and patients with periodontitis (periodontitis) in 5 mM and 25 mM glucose. The dashed line represents the baseline secretion by the untreated coculture model in both 5 mM and 25 mM glucose. Results are the mean ± SEM. One independent replicate and two technical replicates were undertaken for each condition. Two-way ANOVA with the Tukey *post hoc* test. * *p* < 0.05, ** *p* < 0.01, **** *p* < 0.0001.

**Figure 2 ijms-24-08832-f002:**
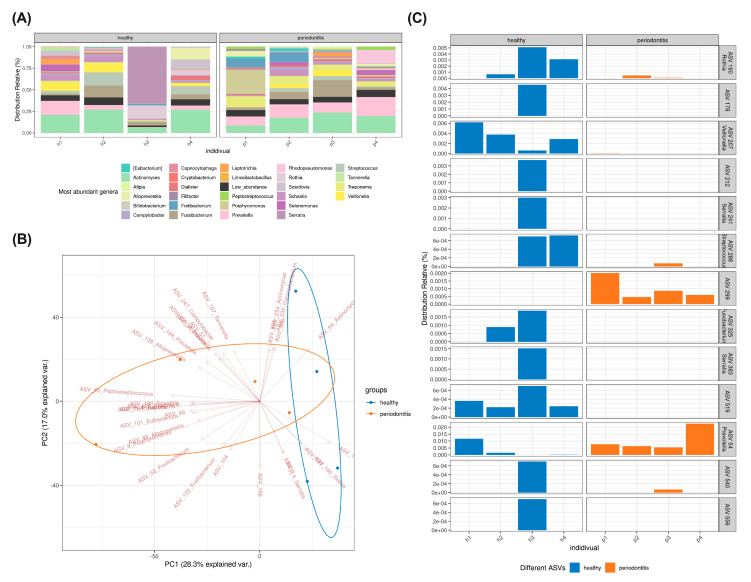
The subgingival microbiome highlights the enrichment of particular amplicon sequence variants (ASVs). (**A**) Relative abundance of most abundant genera following 16s rRNA gene sequencing of subgingival microbiome from healthy participants and patients with periodontitis. (**B**) Principal component analysis (PCA) of the ASV analysis of microbiomes from healthy participants and patients with periodontitis. (**C**) Enrichment of ASV of microbiomes from healthy participants and patients with periodontitis. Samples from healthy donors are identified h1 to h4 while the samples from patients with periodontitis are identified p1 to p4.

**Figure 3 ijms-24-08832-f003:**
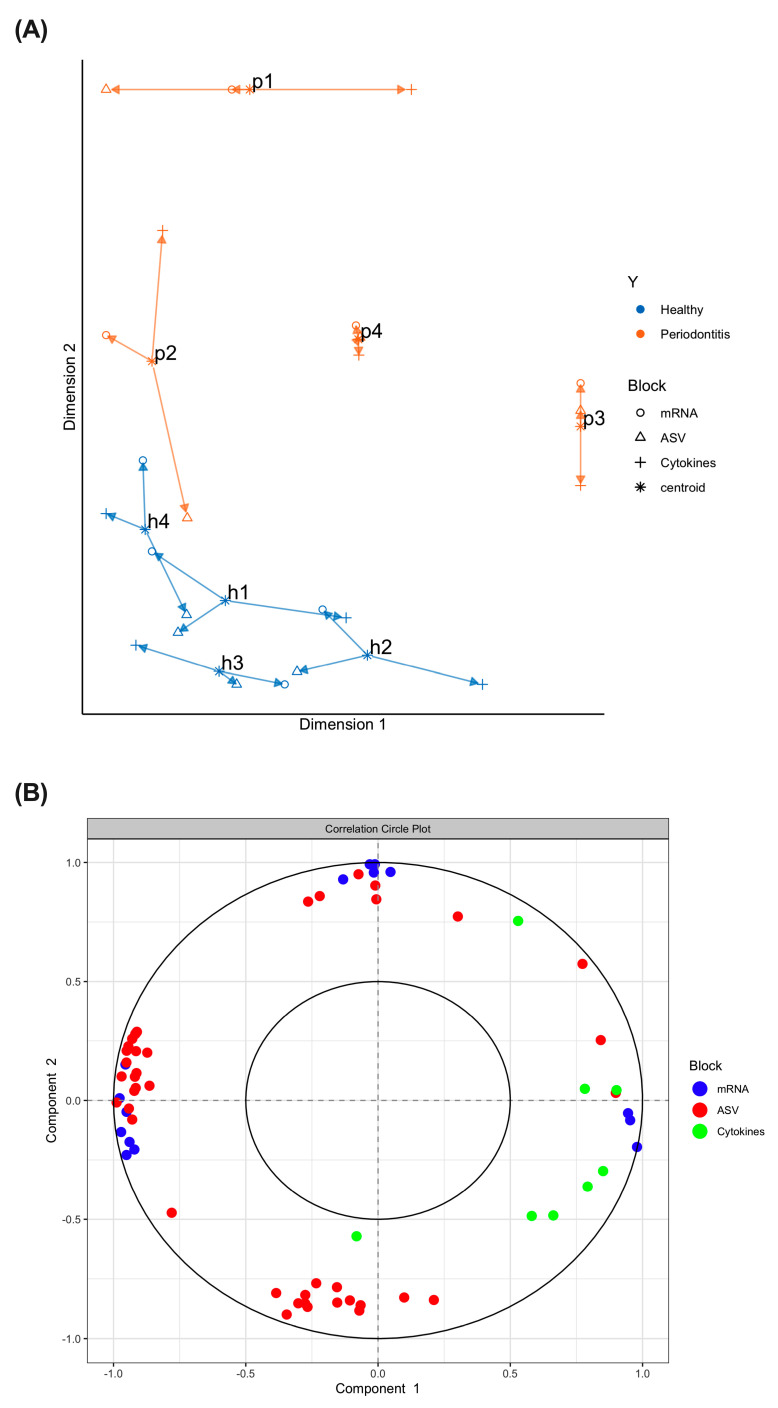
Multi-omics correlations by computational biology. (**A**) Arrow plot. This plot is the superposition of 3 PCAs corresponding to the 3 layers of omics data. For each sample, one arrow represents a set of omics data and the mean of the 3 layers is the centroid. If the arrows are close, the 3 layers of data are similar. On the contrary, if the arrows are apart, then the 3 layers are different. (**B**) Varplot. Positively correlated variables are grouped together. Negatively correlated or anti-correlated variables are positioned on opposite sides of the plot origin (opposed quadrants). Samples from healthy donors are identified h1 to h4 while the samples from patients with periodontitis are identified p1 to p4.

**Figure 4 ijms-24-08832-f004:**
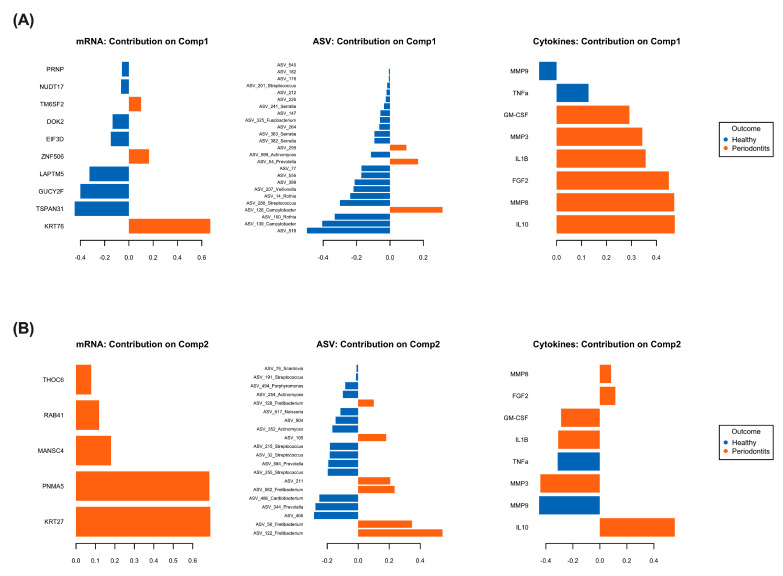
Multi-omics correlations by computational biology analyses reveal key variables contributing to periodontitis-induced inflammatory response in a hyperglycemic microenvironment. (**A**) Projection of the most important variables contributing to the varplot’s component 1. (**B**) Projection of the most important variables contributing to the varplot’s component 2.

## Data Availability

Data are contained within the article.

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
