# Peer review of "Multi-Omics Data Integration Reveals Key Variables Contributing to Subgingival Microbiome Dysbiosis-Induced Inflammatory Response in a Hyperglycemic Microenvironment"

_ijms, 2023, doi:10.3390/ijms24108832_

Round 1
Reviewer 1 Report
In this manuscript, Sarah Lafleur and colleagues, investigate the impact of a hyperglycemic microenvironment on the inflammatory response and transcriptome of a gingival coculture model stimulated with dysbiotic subgingival microbiomes. They used HGF-1 cells overlaid with U937 macrophage-likes cells stimulated with subgingival microbiome collected from four healthy donors and four patients with periodontitis. Through a multi-omics integration analysis, they found unique interrelated bacterial genera, genes, and pro-inflammatory cytokines involved in the regulation of the inflammatory response in a hyperglycemic microenvironment. The manuscript is very interesting, I only have a few questions raise to the authors to improve the quality of the work.
Why did the authors decide to study IL-10, TNF-A, IL-1B, and IL-6 as cytokines with pro-inflammatory activity, but not others?
Periodontitis is an irreversible chronic inflammatory disease that leads to tissue destruction caused by the production of inflammatory mediators including cytokines and matrix metalloproteinases (MMPs) and generates oxidative stress by the excessive production of free radicals creating an inflamed environment. Did the authors evaluate oxidative stress in their experimental model?
I advise the authors to increase the introduction section, especially for the correlation between periodontitis and type 2 diabetes
The purpose of the work should be clearer, and I advise the authors to emphasize this in the discussion section as well.
The authors should specify acronyms in full.
Figures 2, 3, and 4 should be sharp.
Additional materials should be incorporated in a separate file but not in the manuscript
In my opinion, despite the limitations set out by the authors, the manuscript is interesting and suitable for publication after these revisions. It fits perfectly into the field of the journal. The bibliography is also quite complete, perhaps I would add a few more recent manuscripts, but overall it is fine. The conclusions are perfectly supported by the cited manuscripts and their comments.
Minor editing of English language required
Author Response
In this manuscript, Sarah Lafleur and colleagues, investigate the impact of a hyperglycemic microenvironment on the inflammatory response and transcriptome of a gingival coculture model stimulated with dysbiotic subgingival microbiomes. They used HGF-1 cells overlaid with U937 macrophage-likes cells stimulated with subgingival microbiome collected from four healthy donors and four patients with periodontitis. Through a multi-omics integration analysis, they found unique interrelated bacterial genera, genes, and pro-inflammatory cytokines involved in the regulation of the inflammatory response in a hyperglycemic microenvironment. The manuscript is very interesting, I only have a few questions raise to the authors to improve the quality of the work.
Why did the authors decide to study IL-10, TNF-A, IL-1B, and IL-6 as cytokines with pro-inflammatory activity, but not others?
- Due to the limited quantity of supernatants from our experimental cell culture model, we have chosen to study pro-inflammatory cytokines secretion by using a Luminex assay. A total of 15 cytokines and MMPs have been screening simultaneously. These cytokines were chosen based on the literature for HGF-1 and U937 inflammatory responses (IL-6, IL-8, CXCL10, C5a, IFNg, MCP1, IL-18, IL-10, TNFa, GM-CSF, IL-1B, FGF2, MMP8, MMP3 and MMP9). Out of these 15 pro-inflammatory cytokines measured, 6 were out of the assay standard curve detection limits. We were thus unable to quantify MCP1, IL-18, C5a, IFNg, CXCL10 and IL-8. The results for the other 9 cytokines are included in the manuscript.
Periodontitis is an irreversible chronic inflammatory disease that leads to tissue destruction caused by the production of inflammatory mediators including cytokines and matrix metalloproteinases (MMPs) and generates oxidative stress by the excessive production of free radicals creating an inflamed environment. Did the authors evaluate oxidative stress in their experimental model?
- We thank the reviewer for this comment. Unfortunately, we did not measure oxidative stress in our experimental model. We agree that ROS are important contributors to the inflamed environment, and we have added sentences on this topic in the discussion.
I advise the authors to increase the introduction section, especially for the correlation between periodontitis and type 2 diabetes
- We thank the reviewer for this comment. The introduction has been updated accordingly.
The purpose of the work should be clearer, and I advise the authors to emphasize this in the discussion section as well.
- We thank the reviewer for this comment. Introduction and discussion have been updated accordingly.
The authors should specify acronyms in full.
- Acronyms not in full name have been updated.
Figures 2, 3, and 4 should be sharp.
- We thank the reviewer for this comment. The quality of the figures has been improved and the figures downloaded in the revised manuscript.
Additional materials should be incorporated in a separate file but not in the manuscript.
- The additional material has been incorporated in its appropriate section by the MDPI assistant editor during the manuscript formatting.
In my opinion, despite the limitations set out by the authors, the manuscript is interesting and suitable for publication after these revisions. It fits perfectly into the field of the journal. The bibliography is also quite complete, perhaps I would add a few more recent manuscripts, but overall it is fine. The conclusions are perfectly supported by the cited manuscripts and their comments.
- We thank the reviewer for this comment. We have enhanced the introduction and discussion sections. The bibliography has thus been updated accordingly.
Reviewer 2 Report
The correlation between diabetes and periodontal disease is a very current topic in research. However, the effects of hyperglycemic microenvironment on the host-microbiome interactions and host inflammatory response during periodontitis are currently less known. The current researcher succeeded, through standardized high-performance technological methods, to investigate the impacts of a hyperglycemic microenvironment on the inflammatory response and transcriptome of a gingival coculture model stimulated with dysbiotic subgingival microbiome. It is an exceptional research because the authors used an advanced multi-omics bioinformatic data integration model for data analysis. The working methods are clearly exposed, the statistical analysis is the performance, the discussions are extensive, the bibliography is exhaustive with the latest articles on this topic. The work is original and for the first time in the literature it demonstrated the following: the multi-omics integration analysis unveiled complex interrelationships involved in the regulation of periodontal inflammation in response to hyperglycemic microenvironment.
Author Response
The correlation between diabetes and periodontal disease is a very current topic in research. However, the effects of hyperglycemic microenvironment on the host-microbiome interactions and host inflammatory response during periodontitis are currently less known. The current researcher succeeded, through standardized high-performance technological methods, to investigate the impacts of a hyperglycemic microenvironment on the inflammatory response and transcriptome of a gingival coculture model stimulated with dysbiotic subgingival microbiome. It is an exceptional research because the authors used an advanced multi-omics bioinformatic data integration model for data analysis. The working methods are clearly exposed, the statistical analysis is the performance, the discussions are extensive, the bibliography is exhaustive with the latest articles on this topic. The work is original and for the first time in the literature it demonstrated the following: the multi-omics integration analysis unveiled complex interrelationships involved in the regulation of periodontal inflammation in response to hyperglycemic microenvironment.
- We thank the reviewer for these positive comments.
Reviewer 3 Report
Manuscript of considerable interest for the dental sector, a very topical topic when looking for the
as possible to take proactive action to maintain the tissue integrity of the gingival margin
Abstract to highlight the results obtained
Sufficient and inherent keywords, check that they are registered on MeSH
Introduction: add the new classification of periodontal disease associated with systemic pathologies
Results: images not in high resolution, graphics that are not easy to understand, try to add hypertext references
Discussion: add as a future goal the use of natural substances such as probiotics, paraprobiotics, lasers and ozone to reduce the bacterial load and fight dysbiosis as already studied by Prof Scribante's research group
Materials and Methods: Well described
Conclusions: add proactive action
bibliography: add required references
Edit grammatical errors and scientific terms
Author Response
Manuscript of considerable interest for the dental sector, a very topical topic when looking for the
as possible to take proactive action to maintain the tissue integrity of the gingival margin
Abstract to highlight the results obtained
Sufficient and inherent keywords, check that they are registered on MeSH
- We thank the reviewer for this comment. The keywords have been checked on MeSH.
Introduction: add the new classification of periodontal disease associated with systemic pathologies
- We thank the reviewer for this comment. The new classification has been added in the introduction section.
Results: images not in high resolution, graphics that are not easy to understand, try to add hypertext references
- We thank the reviewer for this comment. The quality of the figures has been improved and the figures downloaded in the revised manuscript.
Discussion: add as a future goal the use of natural substances such as probiotics, paraprobiotics, lasers and ozone to reduce the bacterial load and fight dysbiosis as already studied by Prof Scribante's research group
- We thank the reviewer for this comment. Future research goals have been discussed in the discussion.
Materials and Methods: Well described
Conclusions: add proactive action
- We thank the reviewer for this comment. Future research goals have been added in the conclusion.
bibliography: add required references
- We thank the reviewer for this comment. We have enhanced the introduction and discussion sections. The bibliography has thus been updated accordingly.
Round 2
Reviewer 3 Report
Manuscript has been reviewed successfully, can be published